# A tissue-intrinsic IL-33/EGF circuit promotes epithelial regeneration after intestinal injury

Marco Calafiore[1,2], Ya-Yuan Fu [1,2], Paola Vinci[1,2], Viktor Arnhold[1,2], Winston Y. Chang[1,2,3], Suze A. Jansen[4,5], Anastasiya Egorova[1,2], Shuichiro Takashima [1,2,6], Jason Kuttiyara[1,2], Takahiro Ito[1,2], Jonathan Serody [7], Susumu Nakae[8], Heth Turnquist [9], Johan van Es[10], Hans Clevers [5,10,13], Caroline A. Lindemans[4,5], Bruce R. Blazar[11,14] & Alan M. Hanash [1,2,3,12,14] ✉

Intestinal stem cells (ISCs) maintain the epithelial lining of the intestines, but mechanisms regulating ISCs and their niche after damage remain poorly understood. Utilizing radiation injury to model intestinal pathology, we report here that the Interleukin-33 (IL-33)/ST2 axis, an immunomodulatory pathway monitored clinically as an intestinal injury biomarker, regulates intrinsic epithelial regeneration by inducing production of epidermal growth factor (EGF). Three-dimensional imaging and lineage-specific RiboTag induction within the stem cell compartment indicated that ISCs expressed IL-33 in response to radiation injury. Neighboring Paneth cells responded to IL-33 by augmenting production of EGF, which promoted ISC recovery and epithelial regeneration. These findings reveal an unknown pathway of niche regulation and crypt regeneration whereby the niche responds dynamically upon injury and the stem cells orchestrate regeneration by regulating their niche. This regenerative circuit also highlights the breadth of IL-33 activity beyond immunomodulation and the therapeutic potential of EGF administration for treatment of intestinal injury.

The mucosal epithelium of the gastrointestinal (GI) tract consists of a single layer of cells whose functions include absorption of nutrients and containment of enteric microbes. However, the constitutively active mitotic cells of the small intestine (SI) epithelium are highly susceptible to injury from ionizing radiation[1–3]. Interleukin-33 (IL-33) is an alarmin released upon tissue damage, and it can induce differentiation and cytokine production in target cells expressing its receptor, a heterodimer of membrane-bound ST2 (mST2) and IL-1R3[4]. This distress signal is typically thought to target lymphocytes such as T cells and innate lymphoid cells (ILCs)[5–7].

In addition to the release of IL-33, intestinal tissue damage has been associated with elevated levels of soluble ST2 (sST2), which can

[1]Human Oncology & Pathogenesis Program, Memorial Sloan Kettering Cancer Center, New York, NY 10065, USA. [2]Department of Medicine, Memorial Sloan Kettering Cancer Center, New York, NY 10065, USA. [3]Immunology and Microbial Pathogenesis Program, Weill Cornell Medical College, New York, NY 10065, USA. [4]Division of Pediatrics, Regenerative Medicine Center, University Medical Center Utrecht, Utrecht University, 3508 AB Utrecht, Netherlands. [5]Princess Máxima Center for Pediatric Oncology, 3584 CS Utrecht, Netherlands. [6]Department of Hematology, National Hospital Organization Kyushu Medical Center, Fukuoka, Fukuoka 810-8563, Japan. [7]Department of Microbiology and Immunology, Lineberger Comprehensive Cancer Center, University of North Carolina, Chapel Hill, NC, USA. [8]Graduate School of Integrated Sciences for Life, Hiroshima University, Higashi-Hiroshima City, Hiroshima 739-0046, Japan. [9]Starzl Transplantation Institute, Department of Surgery, and Department of Immunology, University of Pittsburgh School of Medicine, Pittsburgh, PA, USA. [10]Hubrecht Institute, Royal Netherlands Academy of Arts and Sciences (KNAW), 3584 CT Utrecht, the Netherlands. [11]Department of Pediatrics, Division of Blood & Marrow Transplant & Cellular Therapy, University of Minnesota, Minneapolis, MN 55455, USA. [12]Department of Medicine, Weill Cornell Medical College, New York, NY 10065, USA. [13]Present address: Roche Pharma Research and Early Development, Basel, Switzerland. [14]These authors contributed equally: Bruce R. Blazar, Alan M. Hanash. ✉e-mail: hanasha@mskcc.org

act as a negative regulator by neutralizing IL-33[8]. Clinically, sST2 levels have been utilized as a predictive biomarker in several pathologic settings such as graft vs. host disease[9,10] and inflammatory bowel disease[11,12]. While much work has focused on the immunomodulatory functions of IL-33 and ST2[5], their tissue-intrinsic roles remain poorly understood. We thus undertook an evaluation of IL-33 biology in homeostasis, radiation injury, and regeneration of the GI tract. Here, we report that the IL-33/ST2 axis mediates cross-talk between ISCs and Paneth cells that promotes epithelial regeneration after damage.

## Results and Discussion

### IL-33 deficiency exacerbates intestinal radiation injury

To evaluate the effects of irradiation on the IL33/ST2 axis, C57BL/6 (B6) mice were exposed to a single 10 Gy dose of total body irradiation (TBI) capable of causing substantial but reversible intestinal injury. IL-33 levels in SI increased shortly after exposure to radiation, peaking at approximately 72 h and then decreasing to near pretreatment levels after five days (Fig. 1a). Similar kinetics were also observed for IL-33 mRNA expression measured by quantitative (q)PCR after TBI (Supplementary Fig. 1a). Dysbiotic changes in the intestinal flora may contribute to cytokine production and inflammation after radiation[13], so mice were pretreated for one week with enrofloxacin and ampicillin[14] to test if host-microbiota interactions might contribute to the IL-33 levels observed after irradiation. However, this antibiotic treatment did not preclude detection of IL-33 after TBI (Supplementary Fig. 1b). We next examined the relationship between intestinal IL-33 and its receptor after irradiation. Levels of the secreted receptor isoform, sST2, and membrane-bound mST2 were measured by qPCR, as protein detection assays do not distinguish the two isoforms within tissues. During the five-day period following TBI, SI crypts isolated from wild-type (WT) mice demonstrated no change in gene expression for the transmembrane receptor mST2 (Supplementary Fig. 1c). In contrast, similar to the measurements of IL-33, sST2 mRNA levels increased shortly after irradiation (Fig. 1b). However, this radiation-induced increase in sST2 expression could not be identified in crypts isolated from IL-33 knockout (KO) mice (Fig. 1b), suggesting that sST2 could be produced in response to IL-33 following radiation injury.

Given the increased intestinal IL-33 expression observed following TBI, we examined the role of endogenous IL-33 in epithelial injury and regeneration within the intestines following irradiation. Comparing WT and IL-33 KO B6 mice, quantification of ileal crypts per circumference did not indicate significant differences in average crypt number due to IL-33 deficiency during homeostasis or three days after TBI, although there was a trend toward reduced crypt number in irradiated IL-33-deficient mice (Supplementary Fig. 1d). Additionally, histologic analysis of crypt depth, an index of mitotic cells and regeneration, indicated crypt elongation three days after TBI, but no difference between the WT and IL-33 KO mice (Supplementary Fig. 1d). Continuing to follow the response to radiation injury, histology was re-examined on day 5 post-TBI (Fig. 1c). Again, unirradiated baseline controls showed no difference between WT and IL-33 KO mice in ileal crypt number (Fig. 1d) or size (Fig. 1e). While WT mice demonstrated a reduction in crypt number five days after irradiation, they also demonstrated typical morphology of regenerative intestinal epithelium with significant crypt hyperplasia (Fig. 1c). Additionally, the average Ki67+ cell frequency in the crypt region was significantly augmented in irradiated WT mice (Fig. 1f, g), further supporting the proliferative regenerating phenotype suggested by their increased crypt depth. In contrast, IL-33-deficient mice demonstrated marked epithelial impairment in response to radiation injury by five days after TBI. Compared to WT ileum, the absence of IL-33 resulted in more severe crypt loss (Fig. 1c, d) and decreased regeneration, as indicated by reduced crypt depth and fewer Ki67+ cells (Fig. 1e–g).

We next examined the stem cell compartment, measuring ISC frequencies using WT and IL-33-deficient Lgr5-LacZ reporter mice. Five

days after TBI, there was a substantially worse reduction of Lgr5+ cells in IL-33 KO mice (Fig. 1h, i), indicating exacerbated radiation-induced loss of ISCs in the absence of IL-33. Consistent with these quantifications, measurement of Lgr5 and Olfm4 mRNA indicated reduced ISC gene expression in crypts from IL-33-deficient mice after TBI (Supplementary Fig. 1e, f). Given that IL-33 can promote differentiation toward secretory cells following bacterial infection[15], we next examined the frequency of Paneth cells, secretory lineage ISC progeny that produce innate antimicrobials and growth factors that can contribute to the stem cell niche[16]. As with the ISCs, the reduction of lysozyme+ Paneth cells five days after TBI (10 Gy) was more severe in IL-33 KO mice (Supplementary Fig. 1g). IL-33 thus increased in the intestines of WT mice after irradiation, and IL-33 contributed to the protection of the ISC compartment and promotion of regeneration following radiation injury.

### Intestinal stems cells express IL-33 after irradiation

In order to investigate sources of IL-33 in the GI tract following radiation injury, intestines from IL-33-GFP reporter mice[17] were evaluated by whole-mount confocal microscopy. Prior to irradiation, three-dimensional projections demonstrated IL-33-GFP expression within the ileal mucosa (Fig. 2a, upper panel). This baseline signal was restricted to the lamina propria, and there were no positive cells within unirradiated crypt epithelium (Fig. 2a, b). Following TBI, there was an increase in ileal IL-33-GFP expression, and GFP+ cells could now be identified within the crypts (Fig. 2a, b). Serial analyses indicated that IL-33-GFP+ crypts could be identified within 24 h after irradiation, that most crypts were GFP+ by 48 h after irradiation, and that the frequency of positive cells within each crypt increased over the course of 72 h (Supplementary Fig. 2a–c).

Evaluating the phenotype of IL-33-expressing cells in the intestines, the GFP signal did not colocalize with CD45 at baseline or after irradiation, indicating that hematopoietic cells were unlikely to be major sources of intestinal IL-33 (Fig. 2c). Within the lamina propria, GFP+ cells also did not express the endothelial marker CD31 (Supplementary Fig. 2d), but the majority did co-express vimentin, indicating that stromal cells such as fibroblasts[15] were likely the main expressors of IL-33 in homeostasis (Supplementary Fig. 2e).

After irradiation, the absence of CD45 colocalization excluded intraepithelial lymphocytes as the source of new IL-33 expression within crypts (Fig. 2c). Examining the epithelium itself, lysozyme-expressing Paneth cells were also negative for GFP (Fig. 2d). However, Paneth cells reside at the crypt base adjacent to the stem cells, and DAPI nuclear staining identified crypt base GFP+ cells after TBI, some of which were in direct contact with GFP−lysozyme+ Paneth cells (Fig. 2d). Indeed, staining for the ISC marker Olfm4[18] colocalized together with GFP expression 48 h after irradiation (Fig. 2e). While not all Olfm4+ cells co-expressed GFP after TBI, Olfm4 staining did colocalize with the vast majority of GFP+ crypt cells, indicating the ISCs to be the primary source of IL-33 expression within crypts in the setting of radiation injury (Fig. 2f).

### IL-33 directly impacts intestinal epithelium

IL-33 can signal to innate and adaptive lymphocytes within tissues[5–7], for example inducing their production of amphiregulin[19,20]. Given this potential to influence pro-regenerative lymphocytes and given that IL-33 deficiency impaired the epithelial regenerative response to radiation injury (Fig. 1), we investigated the role of the immune system in driving IL-33-dependent regeneration by depleting the lymphocytes most likely to be involved. Prior to irradiation (10 Gy TBI), anti-Thy1 or isotype control antibodies were administered to WT and IL-33 KO mice (Supplementary Fig. 3). Flow cytometry confirmed that the anti-Thy1 treatment resulted in substantial depletion of intestinal CD4+ T cells and group 2 ILCs (ILC2s), both of which are known targets of IL-33 (Supplementary Fig. 3a–d). While not typically

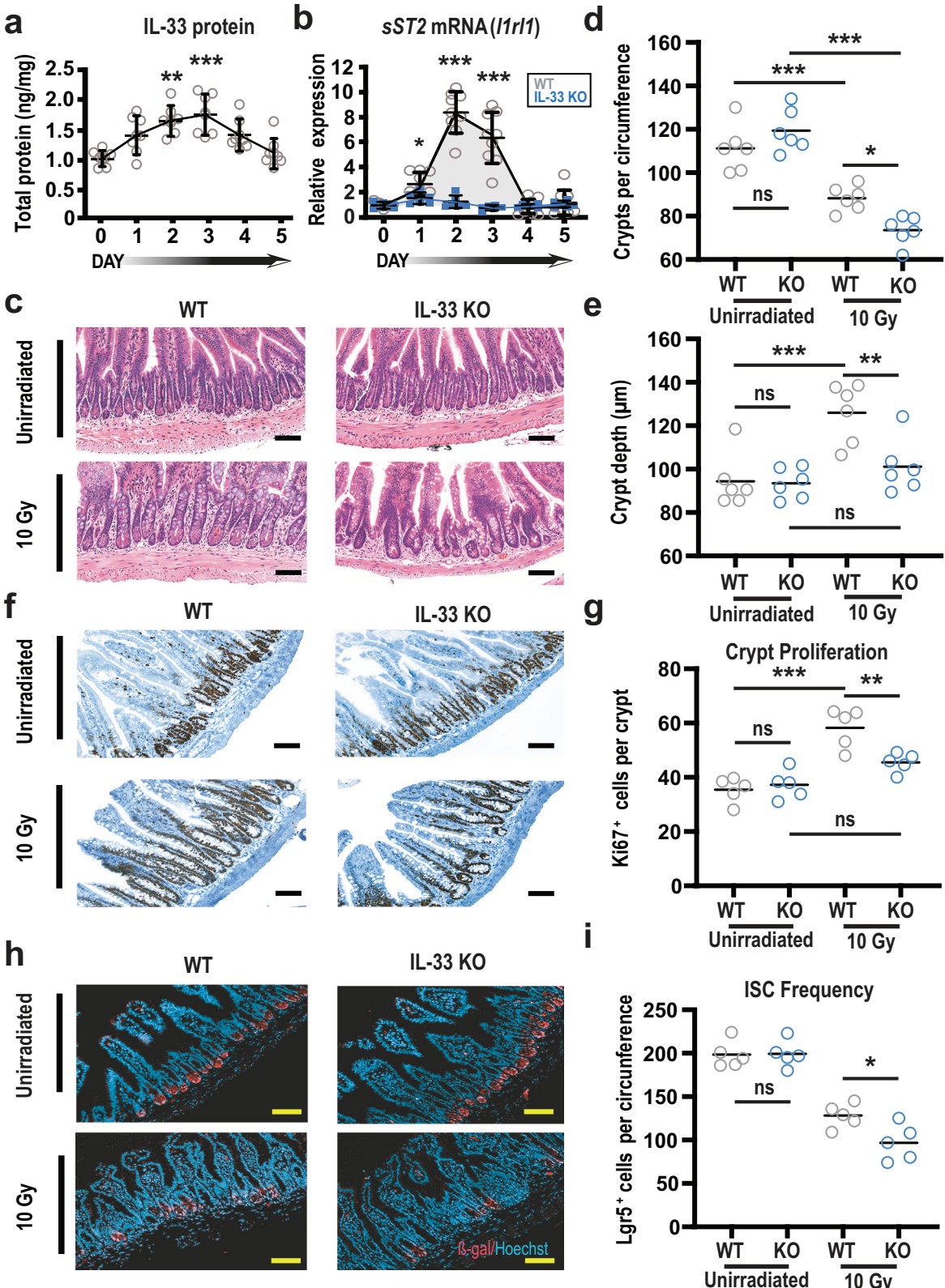

responsive to IL-33, group 3 ILCs (ILC3s) can still be important contributors to immune-mediated intestinal regeneration and ISC regulation[21,22], and ILC3s were also depleted by the anti-Thy1 treatment (Supplementary Fig. 3b, e). The ileum was then examined histologically in the same mice where intestinal lymphocyte frequencies were confirmed by flow cytometry. Even in this immunodepleted setting, IL-33-deficient mice continued to demonstrate more severe crypt loss and reduced crypt size in response to radiation injury compared to WT mice (Supplementary Fig. 3f, g), suggesting that IL-33-dependent regulation of crypt damage and regeneration was not simply a consequence of immunomodulatory alarmin function.

Exploring the possibility of a tissue-intrinsic role for IL-33 in the epithelial response to injury, we utilized an ex vivo organoid culture system to test if IL-33 could influence regeneration by acting directly

**Fig. 1 | IL-33 deficiency increases the severity of intestinal radiation injury and impairs epithelial regeneration. a** Time course of IL-33 protein in small intestine (SI) at baseline and over 5 days after 10 Gy TBI ($n = 7$ mice/group), measured by ELISA and normalized to mg of total protein; data combined from two experiments. **b** Relative expression of soluble IL-33 receptor (sST2) measured by qPCR in SI crypts from WT and IL-33 KO mice following 10 Gy TBI ($n = 9$ WT and $n = 6$ IL-33 KO mice/group); data combined from three independent experiments. **c–e** Terminal SI histology (ileum) in WT and IL-33 KO mice at baseline and 5 days after TBI; shown are representative images (**c**) and quantified analyses of crypt number (**d**) and size (**e**) combined from two experiments ($n = 6$ mice/group). **f, g** Representative images (**f**) and quantification (**g**) of Ki67 immunohistochemistry in ileal crypts at baseline and five days after TBI (10 Gy); data combined from two experiments ($n = 5$ mice/group). **h, i** Representative immunofluorescent images (**h**) and quantification (**i**) of Lgr5-LacZ⁺ cells stained for anti-β-galactosidase (red) and nuclei (blue) in ileum on day 5 after TBI (10 Gy); data combined from two experiments ($n = 5$ mice/group). Statistical analyses were performed using one-way ANOVA multiple comparison testing (**a, d, e, g, i**), or two-tailed Mann-Whitney U (**b**). All dot plots (**a, b, d, e, g, i**) show means, and error bars indicate SEM (**a, b**); *$p < 0.05$, **$p < 0.01$, ***$p < 0.001$; scale bars: 100 μm. Source data for graphs are provided in the Source Data file. The exact *p*-values are as follows: (**a**) comparisons made vs. day 0; day 1, $p = 0.1014$; day 2, $p = 0.0013$; day 3, $p = 0.0002$; day 4, $p = 0.0972$; day 5 $p = 0.9851$; (**b**) WT vs KO comparisons made for each timepoint; day 0, $p = 0.8374$; day 1, $p = 0.0176$; day 2, $p = 0.0004$; day 3, $p = 0.0004$; day 4, $p = 0.9792$; day 5, $p = 0.7544$; (**d**) unirradiated WT vs unirradiated KO, $p = 0.37$; unirradiated WT vs 10 Gy WT, $p = 0.0008$; unirradiated KO vs 10 Gy KO, $p = < 0.0001$; 10 Gy WT vs 10 Gy KO, $p = 0.0343$. (**e**) unirradiated WT vs unirradiated KO, $p = 0.99$; unirradiated WT vs 10 Gy WT, $p = 0.0007$; unirradiated KO vs 10 Gy KO, $p = 0.67$; 10 Gy WT vs 10 Gy KO, $p = 0.007$; (**g**) unirradiated WT vs unirradiated KO, $p = 0.94$; unirradiated WT vs 10 Gy WT, $p = < 0.0001$; unirradiated KO vs 10 Gy KO, $p = 0.11$; 10 Gy WT vs 10 Gy KO, $p = 0.0084$; (**i**) unirradiated WT vs unirradiated KO, $p = 0.99$; 10 Gy WT vs 10 Gy KO 10 Gy $p = 0.039$.

on the epithelium. Although the intestinal epithelium did not express IL-33 at baseline (Fig. 2), imaging of SI IL-33-GFP organoids grown in Matrigel and standard epidermal growth factor (EGF)/Noggin/R-spondin-1 (ENR) media for 48 h indicated that IL-33 expression could be induced within epithelial cells following crypt isolation and ex vivo culture (Fig. 3a). Supporting this observation of IL-33-GFP expression, comparison of WT and IL-33 KO organoids by qPCR also indicated epithelial expression of IL-33 (Fig. 3b), and ex vivo IL-33 expression was elevated within hours of organoid passaging (Supplementary Fig. 4a). Consistent with the identification of ISCs as a source of epithelial IL-33 expression after damage (Fig. 2e, f), supplementation of ENR culture media with CHIR99021 and valproic acid to promote stem cell renewal and increase the proportion of ISCs in culture[23] augmented the frequency of IL-33-GFP⁺ organoids (Supplementary Fig. 4b). Similarly, human intestinal organoids were cultured with either Wnt-conditioned media to promote maintenance of the stem cells in culture or in Wnt-deficient media to promote differentiation and loss of ISCs (Supplementary Fig. 4c, d), and IL-33 expression was substantially greater in the human organoids cultured in conditions maintaining expression of *Lgr5* (Supplementary Fig. 4e).

IL-33 was thus expressed by epithelial cells in organoid cultures, and supplementation of this endogenous IL-33 expression with increasing concentrations of exogenous IL-33 had no apparent impacts on WT organoid number or size, a surrogate marker for ex vivo epithelial proliferation[22] (Supplementary Fig. 5a, b). However, qPCR for the IL-1R3 and mST2 components of the heterodimeric IL-33 receptor complex indicated stable expression and thus the potential for IL-33 to target intestinal organoids (Supplementary Fig. 5c, d). Additionally, culture with exogenous IL-33 led to an increase in mRNA for the secreted IL-33 receptor isoform sST2 (Supplementary Fig. 5e), suggesting that IL-33 may indeed have activity in SI organoids.

Given the expression of IL-33 by epithelial cells in organoid cultures, we studied IL-33's epithelium-intrinsic effects in radiation injury by comparing the sensitivity of WT and IL-33 KO intestinal organoids to irradiation. SI crypt-derived organoids were cultured briefly and then dissociated into single cells to generate a uniform starting population for experimentation. Twenty-four hours after plating as dissociated cells, the developing organoids were irradiated and monitored for survival. Without irradiation, WT and IL-33 KO cultures showed no difference in baseline viability. However, IL-33 KO organoids demonstrated reduced survival compared to WT controls within 48 h of exposure to 6 Gy irradiation (Fig. 3c), and the irradiated KO organoids demonstrated reduced size as well (Fig. 3d). These findings with epithelial cultures thus appeared to recapitulate the in vivo phenotype of IL-33 KO mice with worsened crypt loss and reduced crypt size after TBI (Fig. 1), and, consistent with the effect of IL-33 deficiency in lymphocyte-depleted mice (Supplementary Fig. 3), the irradiated cultures indicated an effect of IL-33 on the epithelial response to radiation injury.

## IL-33 regulates the EGF pathway in intestinal epithelium

Further examining the impacts of IL-33 on epithelial function, IL-33 deficiency was found to increase organoid sensitivity to EGF and its removal from the culture media. Comparing the growth of unirradiated WT and IL-33 KO SI organoids derived from dissociated cells as above, culture for five days in NR media lacking EGF led to substantial reduction in the size of KO organoids (Fig. 3e). This growth impairment after culture in NR media, even in the absence of irradiation, suggested an increased dependence of IL-33-deficient organoids on exogenous EGF. Indeed, the absence of EGF in the culture media was particularly detrimental for IL-33-deficient epithelium, as passaging after culture in NR without EGF resulted in fewer and smaller IL-33 KO organoids compared to WT (Fig. 3f), thus illustrating the importance of EGF supplementation in the setting of IL-33 deficiency.

To better understand the impaired growth of IL-33-deficient epithelial cultures in NR media, we examined their production of EGF. Paneth cells are a known source of EGF for the ISC compartment, and they are a normal constituent of mouse SI organoids[16], so we compared *Egf* expression in WT and IL-33 KO organoid cultures (Fig. 3g). No difference in *Egf* expression was observed when organoids were cultured in EGF-containing ENR media. However, *Egf* was readily detected in WT organoids cultured in EGF-deficient NR media, while expression remained low in IL-33 KO organoids (Fig. 3g), consistent with their increased dependence on exogenous EGF.

Given the reduced *Egf* expression in IL-33 KO organoids, the relationship between IL-33 and EGF was examined more directly. We also investigated the role of p38 MAPK, which is often phosphorylated in response to cell stress[24] and is a known downstream target of the IL-33/ST2 axis in hematopoietic cells[25–28]. Furthermore, western blot analysis showed that IL-33 contributed to p38 phosphorylation in intestinal organoids, as blocking the endogenously produced IL-33 with ST2-Fc reduced the amount of phosphorylated p38 detected in WT organoids (Fig. 3h). Testing the effects of IL-33 and p38 on EGF expression, treatment of IL-33 KO organoid cultures with IL-33 led to increases in EGF mRNA (Fig. 3i) and protein (Fig. 3j), both of which were prevented by the p38 inhibitor SB202190 (Fig. 3i, j). Therefore, inhibition of IL-33 with a soluble form of ST2 reduced phosphorylation of p38, and p38 inhibition blocked IL-33-mediated induction of EGF. All together, these organoid experiments indicated that IL-33 induced epithelial expression of EGF, and they highlighted the impairment of epithelium that is unable to drive its own IL-33-dependent EGF expression.

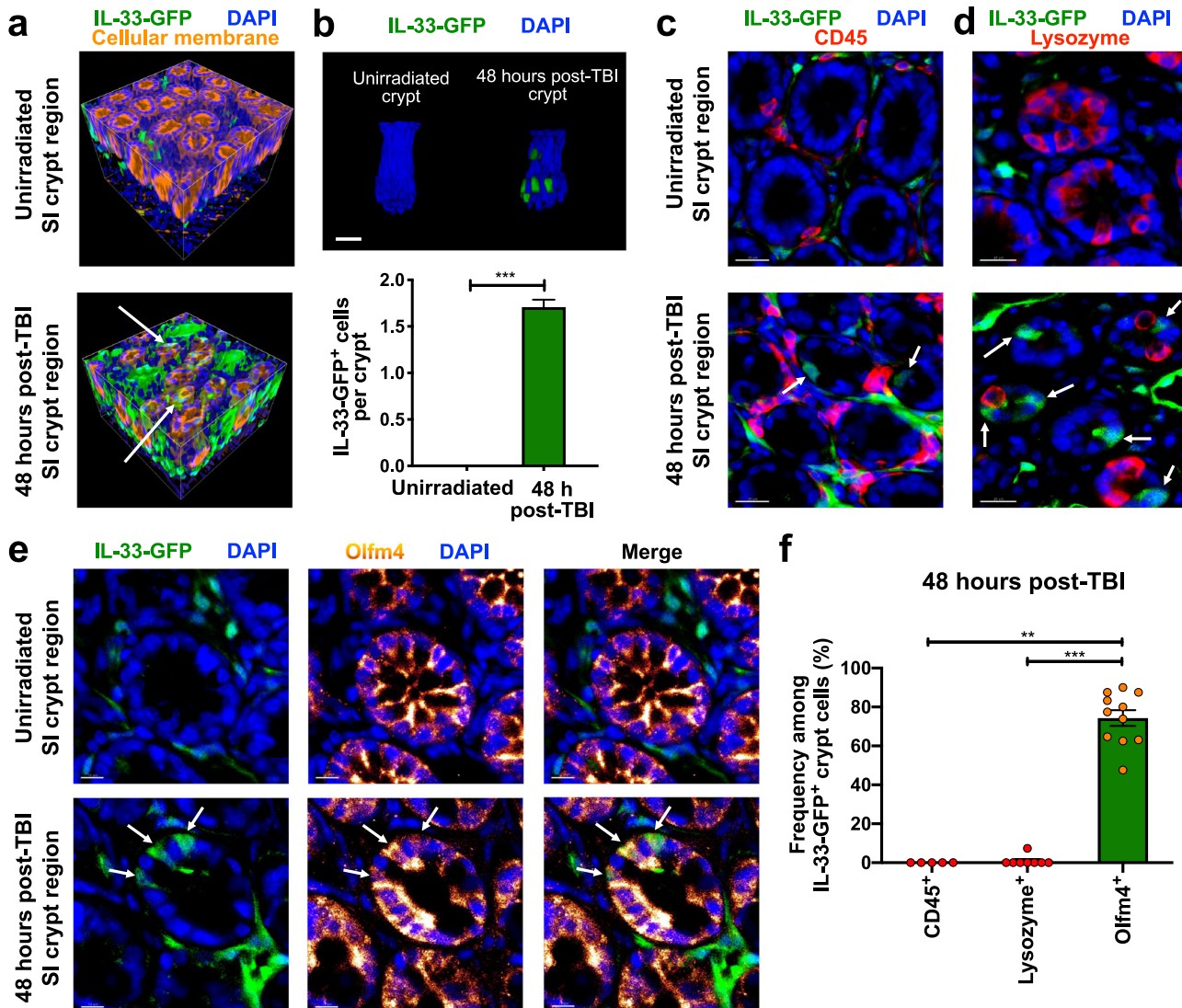

**Fig. 2 | Intestinal stem cells produce IL-33 in response to radiation injury.** Full-thickness small intestine (ileum) from IL-33-GFP reporter mice imaged by three-dimensional (3-D) microscopy. **a** 3-D projections of IL-33-GFP+ cell localization (anti-GFP staining, green) during homeostasis and 48 h after total body irradiation (TBI). Cellular membrane (DiD lipophilic dye staining, orange) and nuclear signals (DAPI staining, blue) indicate tissue architecture. **b** Representative 3-D images of crypts and quantification of IL-33-GFP+ cells at baseline and 48 h after TBI ($n = 379$ crypts/group at baseline, $n = 476$ crypts/group after TBI); scale bar: 25 μm. **c** 2-D optical slices from 3-D imaging of crypt regions with anti-CD45 staining for hematopoietic cells including lymphocytes (red); scale bars: 20 μm. **d** 2-D optical slices from 3-D imaging of crypt regions with staining for Paneth cells (anti-lysozyme, red); scale bars: 20 μm. **e** 2-D optical slices from 3-D imaging of crypt regions with staining for Olfm4+ ISCs (anti-Olfm4, orange glow); scale bars: 10 μm. **f** CD45+, lysozyme+, and Olfm4+ frequencies in IL-33-GFP+ crypt cells 48 h after TBI; $n = 5$ (CD45+), $n = 8$ (lysozyme+), and $n = 11$ (Olfm4+) independent 3-D fields. Arrows in (**a**) and (**c**–**e**) indicate IL33-GFP+ crypt cells. Statistical analyses were performed using a two-tailed Mann-Whitney U test (**b**) or Kruskal-Wallis multiple comparison testing (**f**). Bar graphs show means, and error bars indicate SEM (**b**, **f**); **$p < 0.01$ and ***$p < 0.001$. Source data for graphs are provided in the Source Data file. The exact $p$-values are as follows: (**b**) Unirradiated vs post-TBI, $p < 0.0001$; (**f**) Olfm4+ vs Lysozyme+, $p < 0.0001$; Olfm4+ vs CD45+ $p = 0.0014$.

## Paneth cells increase EGF expression in response to IL-33

We next examined the relationship between IL-33 and EGF in vivo. Assessing the setting of radiation injury, SI *Egf* expression increased five days after TBI in WT mice but not in IL-33 KO mice (Fig. 4a), indicating that IL-33 contributed to EGF gene expression in vivo as well. To localize expression within the mucosa, mRNA was isolated separately from mechanically disrupted villi and from purified crypts. No radiation-induced changes were observed in EGF receptor (EGFR) mRNA expression in villi or crypts from WT mice (Supplementary Fig. 6), and no radiation-induced changes in *Egf* were identified in villi either (Fig. 4b). Rather, the increase in *Egf* expression following irradiation localized to the crypts (Fig. 4b), consistent with the possibility that Paneth cells could be the source.

To further localize mRNA changes within the crypt compartment in vivo, we quantified gene expression in ISCs and Paneth cells using the RiboTag strategy of lineage-specific RNA isolation[29]. Cre-driven RiboTag expression provides targeted hemagglutinin (HA) labeling of ribosomes, which can then be isolated using anti-HA antibodies for analysis of RNA expression from individual populations within tissues. This approach enriches for transcripts actively engaged in translation and avoids transcriptional artifacts induced by extensive manipulation from processing tissues into single cells and then purifying specific populations, thus providing lineage-specific RNA expression more closely reflecting the functionally relevant transcriptome in vivo. RiboTag mice were crossed to express tamoxifen-inducible Cre recombinases in Ofm4+ cells for labeling ISC ribosomes (Olfm4-Ribo) and in Lysozyme-1+ cells to

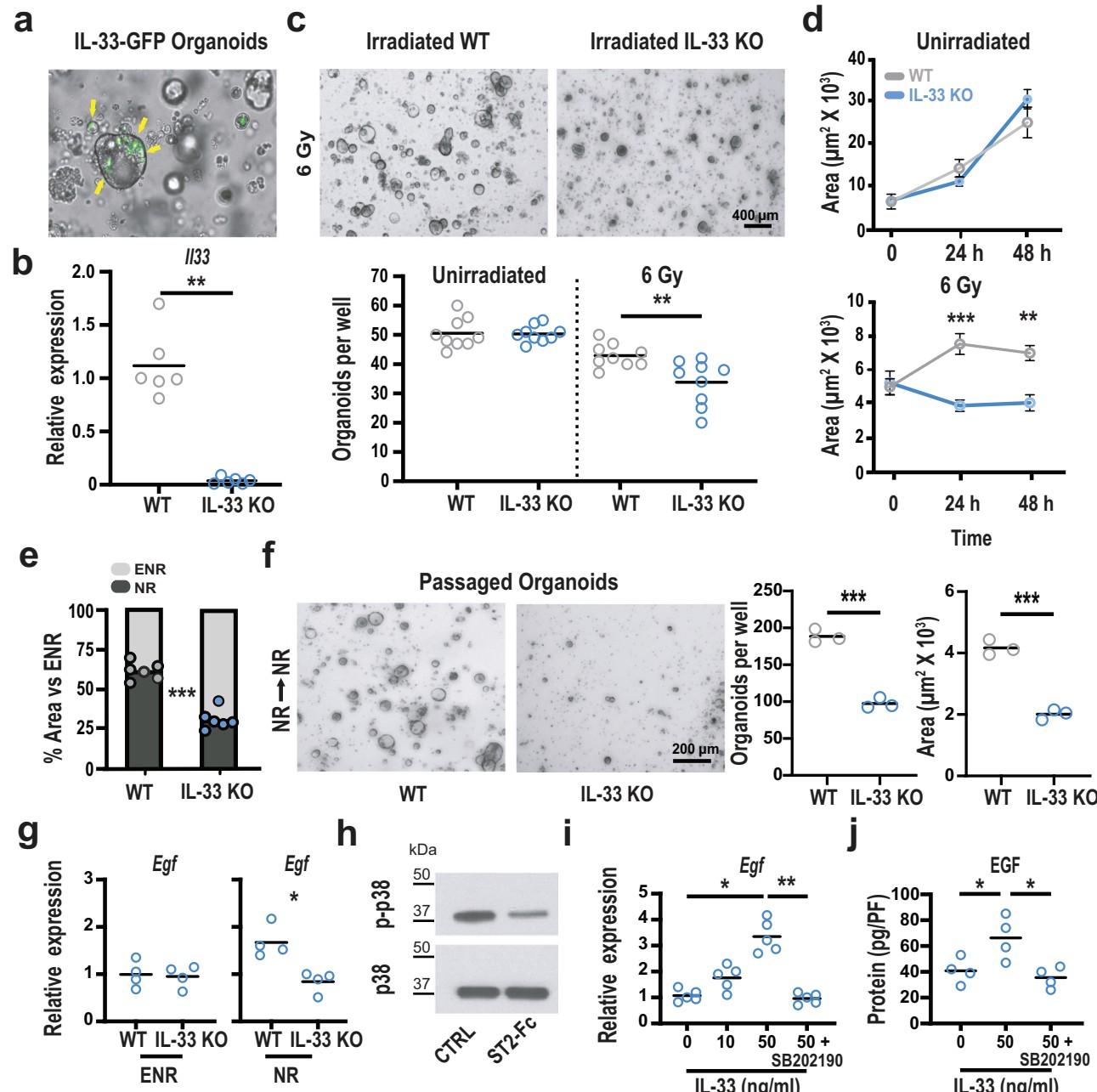

**Fig. 3 | IL-33 regulates EGF production. a** Representative imaging of IL-33-GFP small intestine (SI) organoids cultured ex vivo for 48 hs. Arrows indicate GFP⁺ cells within the organoids. **b**, Relative expression of IL-33 mRNA in WT and IL-33-deficient organoids; data combined from three experiments ($n = 6$ culture wells/group). **c, d** WT and IL-33 KO organoid growth after irradiation. **c** Representative images and quantification of viable WT and IL-33 KO organoids 48 h after irradiation ex vivo; combined from three independent experiments ($n = 9$ wells/group). **d** Time course of WT (gray) and IL-33 KO (blue) organoid size following irradiation ($n = 3$ mice/group). **e** Size of organoids cultured in EGF-deficient NR media relative to their growth in EGF-containing ENR media ($n = 6$ wells/group). **f** Passaged WT and IL-33 KO SI organoids, cultured for five days in NR media without EGF before and after passaging; $n = 3$ mice/group from three independent experiments. **g** qPCR analysis of Egf expression in WT and IL-33 KO organoids cultured in ENR or NR media for four days; data combined from two independent experiments ($n = 4$ mice/group). **h** Representative western blot images of total p38 and phosphorylated p38 (p-p38) from organoids cultured +/− ST2-Fc (2 μg/ml) for eight hours (representative of 3 independent experiments). **i** qPCR analysis of Egf mRNA

expression in IL-33 KO organoids cultured with IL-33 in NR media for 24 h +/− p38 inhibitor SB202190; $n = 5$ wells/group, combined from two experiments. **j** ELISA for EGF protein in homogenized IL-33 KO organoids cultured with IL-33 in NR media for 24 h +/− SB202190; protein level normalized for cellular content by DNA picogreen fluorescence (PF); $n = 4$ wells/group, combined from two experiments. Comparisons were performed using two-tailed unpaired $t$-tests (**c, f**) two-tailed Mann-Whitney U tests (**b, g**), Kruskal-Wallis multiple comparison testing (**i**), or ANOVA multiple comparison testing (**d, j**). Graphs indicate the mean, and error bars (**d**) indicate SEM; *$p < .05$, **$p < 0.01$, ***$p < 0.001$. Source data for graphs and the full blot for the cropped image in (**h**) are provided in the Source Data file. The exact $p$-values are as follows: (**b**) WT vs KO $p = 0.022$; (**c**) unirradiated WT vs unirradiated KO, $p = 0.91$; 6 Gy WT vs 6 Gy KO, $p = 0.006$; (**d**) 6 Gy WT vs KO at 24 h, $p < 0.0008$; 6 Gy WT vs KO at 48 h, $p = 0.0015$; (**e**) WT vs KO in NR, $p < 0.0001$; (**f**) organoids/ well, WT vs KO, $p = 0.0002$; area, WT vs KO, $p = 0.0002$; (**g**) ENR, WT vs KO, $p > 0.99$; NR, WT vs KO, $p = 0.029$; (**i**) 0 vs 50, $p = 0.0137$; 50 vs 50 + SB202190, $p = 0.0033$; (**j**) 0 vs 50, $p = 0.04$; 50 vs 50 + SB202190, $p = 0.015$.

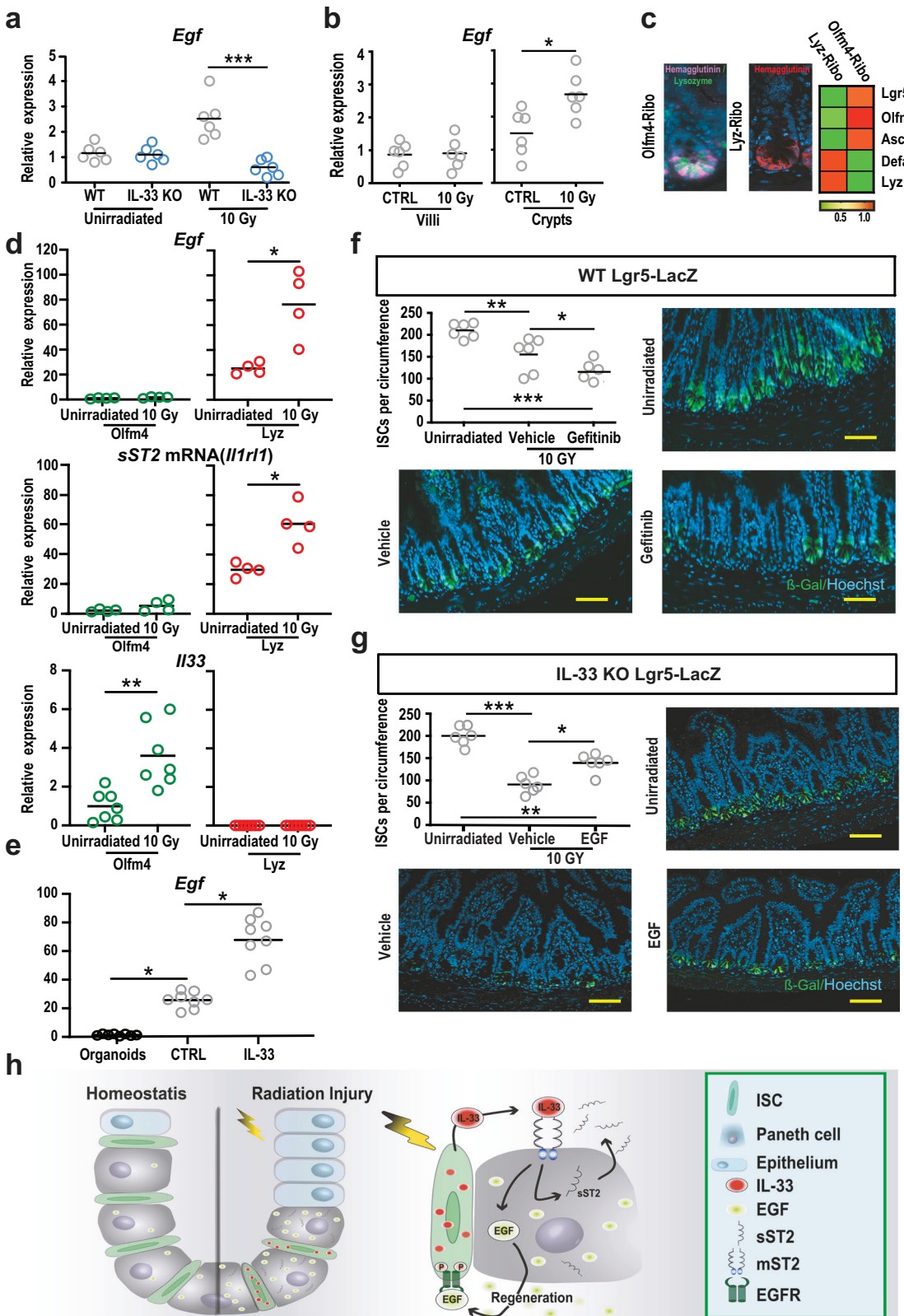

label Paneth cell ribosomes (Lyz-Ribo). For Olfm4, tamoxifen-induced labeling shortly before purification restricted the HA tag to ISCs without labeling of stem cell progeny, whereas for lysozyme, tamoxifen could be administered several days in advance to maximize ribosome labeling in terminally differentiated Paneth cells. Lineage-specific tagging in vivo was validated by both immuno-fluorescent histology and qPCR (Fig. 4c).

We next utilized the RiboTag system to compare proteome-relevant gene expression within the ISC compartment at baseline and during the early stages of radiation injury. SI was harvested 48 h after TBI (10 Gy), the RNA attached to HA-tagged ribosomes was isolated, and qPCR was performed. Within immunoprecipitated mRNA transcripts from Lyz-Ribo mice, *Egf* mRNA was present at baseline and increased following TBI (Fig. 4d), confirming upregulation of EGF

**Fig. 4 | Paneth cells respond to IL-33 by promoting regeneration with EGF. a** *Egf* expression in WT and IL-33 KO mouse small intestine (SI) at baseline and five days after TBI (10 Gy), measured by qPCR (*n* = 6 mice/group, combined from two independent experiments). **b** Villous and crypt expression of *Egf* at baseline and five days after TBI (*n* = 6 mice/group, combined from two experiments). **c** RiboTag hemagglutinin (HA) lineage labeling. Left: immunofluorescence of HA+ cells 20 h (Olfm4-Ribo) or 5 days (Lyz-Ribo) after tamoxifen injection in vivo; anti-HA staining in pink for Olfm4-Ribo (with anti-lysozyme co-stain in green) and anti-HA staining in red for Lyz-Ribo; representative of staining from 5 mice. Right: qPCR profiling of the ISC compartment using anti-HA immunoprecipitates from RiboTag mice. **d** qPCR of RiboTag-isolated mRNA from ISCs and Paneth cells at baseline and five days after TBI (10 Gy); *Egf*, *n* = 4 mice/group from two experiments; *Il1rl1* (sST2), *n* = 4 mice/group from two experiments; *Il33*, *n* = 7 mice/group from three experiments. **e** qPCR for *Egf* in SI organoids and in sort-purified Paneth cells +/− exposure to IL-33 (*n* = 8 wells/group). **f, g** Quantification and imaging of Lgr5-LacZ+ ISCs in ileum five days after TBI (10 Gy). Representative images show anti-β-gal immunofluorescent staining (green) and DAPI nuclear stain (blue) five days after TBI (10 Gy). **f** WT reporter mice +/− gefitinib treatment (1 mg/mouse) or vehicle (daily for three days starting the day of irradiation). Data combined from two experiments (n = 6 mice/ group). **g** IL-33 KO reporter mice +/− treatment with EGF (10 μg/mouse) or PBS

(daily for three days starting the day of irradiation). Data combined from two experiments (*n* = 6 mice/group). **h** Proposed model of IL-33-mediated regulation of regeneration within the ISC compartment after radiation injury: IL-33 was produced by epithelial stem cells after irradiation, and Paneth cells responded to IL-33 by increasing production of EGF, driving epithelial proliferation, crypt maintenance, and ISC recovery after damage. IL-33 also induced a negative feedback response, promoting expression of its own negative regulator sST2. Comparisons performed using a two-tailed Mann-Whitney U test (**b**, **d**), Kruskal-Wallis multiple comparison testing (**a**, **e**), or ANOVA multiple comparison testing (**f**, **g**); *p < 0.05, **p < 0.01, ***p < 0.001; scale bars: 100 μm. Source data for graphs are provided in the Source Data file. The exact p values are as follows: (**a**) WT vs KO, 10 Gy, *p* = 0.0003; (**b**) Villi, CTRL vs 10 Gy, *p* = 0.98; crypts, CTRL vs 10 Gy, *p* = 0.026; (**d**) *Egf*, Olfm4 Unirradiated vs 10 Gy, *p* = 0.17; *Egf*, Lyz Unirradiated vs 10 Gy, *p* = 0.026; *sST2*, Olfm4 Unirradiated vs 10 Gy, *p* = 0.23; *sST2*, Lyz Unirradiated vs 10 Gy, *p* = 0.029; *Il33*, Olfm4 Unirradiated vs 10 Gy, *p* = 0.0012; *Il33*, Lyz Unirradiated vs 10 Gy, *p* > 0.99; (**e**) CTRL vs Organoids, *p* = 0.047; CTRL vs IL-33, *p* = 0.047; (**f**) Unirradiated vs Vehicle, *p* = 0.0092; Unirradiated vs Gefinitib, *p* = < 0.0001; Vehicle vs Gefinitib, *p* = 0.03; (**g**) Unirradiated vs Vehicle, *p* = < 0.0001; Unirradiated vs EGF, p = 0.0022; Vehicle vs EGF, *p* = 0.011.

production by Paneth cells in the setting of radiation injury. Paneth cells were also found to increase mRNA expression of sST2, the negative regulator of IL-33, while no Paneth cell expression of IL-33 message was detected (Fig. 4d). In contrast, immunoprecipitated RNA from Olfm4-Ribo mice revealed little expression of EGF or sST2 before or after TBI but demonstrated notable upregulation of IL-33 in response to irradiation (Fig. 4d). Therefore, consistent with the observations from three-dimensional imaging (Fig. 2), ISCs could be potent sources of IL-33 adjacent to Paneth cells following radiation injury, and, considering the finding of IL-33-induced EGF expression (Fig. 3), Paneth cells had the potential to respond to IL-33 by promoting regeneration with EGF and providing negative feedback to the system via the endogenous IL-33 inhibitor sST2.

To directly investigate the potential of IL-33 to augment Paneth cell production of EGF, Lysozyme+CD24hi Paneth cells[16] from Lysozyme-DsRed reporter mice were purified by cell sorting (Supplementary Fig. 7a). Evaluation by qPCR confirmed enrichment of lysozyme expression and depletion of Lgr5 (Supplementary Fig. 7b, c). Consistent with the imaging and RiboTag findings, qPCR also indicated that sorted Paneth cells expressed little mRNA for IL-33 (Supplementary Fig. 7d). Moreover, sort-purifying Paneth cells enriched for EGF gene expression in comparison to organoids, and exposure to rmIL-33 (50 ng/ml) for 24 h was sufficient to induce purified Paneth cells to increase expression of *Egf* (Fig. 4e).

Previous work has identified epithelial IL-33 expression downstream of Yap activation in the setting of intestinal radiation injury[30,31]. However, such studies have not observed IL-33 expression in typical crypt base columnar ISCs expressing Lgr5 and Olfm4 mRNA. To better understand our identification of IL-33 expression in Olfm4+ cells following radiation injury, we first examined crypt Olfm4 mRNA expression in the first days after TBI (10 Gy). Its expression appeared stable 24 h after irradiation, but by 48 h the amount of Olfm4 message had reduced substantially (Supplementary Fig. 8a). Treating Olfm4-Ribo mice with tamoxifen to label ribosomes with HA 24 h post-TBI, when Olfm4 expression remained intact, resulted in persistent HA labeling of the crypt base by 48 h post-TBI (Supplementary Fig. 8b), despite the reduction in Olfm4 gene expression at this timepoint. Additional anti-HA immunostaining in the colon, where Olfm4 is not expressed in mice, confirmed an absence of HA in this tissue, demonstrating the specificity of the HA label (Supplementary Fig. 8c). These data thus indicated that although ISC gene expression was reduced 48 h after 10 Gy TBI, at least some of the cells that had expressed Olfm4 remained present at the crypt base. While the reduction in gene expression for ISC markers may limit their detection in the setting of radiation injury,

use of the RiboTag system, which irreversibly labeled the crypt base stem cells, facilitated identification of the cells persisting after irradiation and evaluation of some of their functionally relevant transcripts such as IL-33.

## EGF treatment circumvents IL-33 deficiency in vivo

Finally, given the impaired regeneration in IL-33 KO mice after TBI, the effects of irradiation on IL-33 expression, and the IL-33-dependent regulation of EGF, we sought to further examine the mechanistic relationship between IL-33 and EGF in epithelial regeneration after radiation injury. To investigate the role of the EGF/EGFR axis and test if the EGF deficiency observed in IL-33 KO mice could be responsible for their impaired regeneration, WT Lgr5-LacZ mice were treated with the EGFR inhibitor gefitinib or vehicle by oral gavage daily for three days starting immediately after TBI. Histologic examination on day 5 following the irradiation indicated that the loss of Lgr5-LacZ+ ISCs was more severe in gefitinib-treated mice (Fig. 4f). EGFR inhibition also led to fewer Ki67+ crypt cells and fewer crypts overall (Supplementary Fig. 9a, b), thereby appearing to phenocopy the increased epithelial sensitivity to TBI and impaired intestinal regeneration observed in both IL-33-deficient and Paneth-cell-deficient mice. Unsurprisingly, EGFR inhibition was also highly toxic for regenerating epithelium ex vivo, as gefitinib exposure prevented organoid growth from WT crypt epithelial cells (Supplementary Fig. 9c).

Next, given that IL-33 deficiency led to reduced EGF expression, we tested if intraperitoneal administration of exogenous EGF was sufficient to limit the radiation sensitivity of IL-33-deficient mice and restore regeneration. Indeed, EGF treatment enhanced crypt Lgr5-LacZ frequencies (Fig. 4g) and increased crypt depth and proliferating Ki67+ cells (Supplementary Fig. 9d, e) in irradiated IL-33 KO mice. Therefore, IL-33 induced expression of EGF, the EGF pathway contributed to intestinal regeneration after TBI, and the defective regenerative response in IL-33-deficient mice could be improved by administration of exogenous EGF.

In conclusion, IL-33 coordinated epithelial regeneration following intestinal radiation injury, limiting crypt loss and promoting recovery of the stem cell compartment. ISCs expressed IL-33 following irradiation, and adjacent Paneth cells responded by upregulating expression of EGF. Given their close proximity, ISCs sit in an optimal position to transmit their IL-33 signal to neighboring Paneth cells. As such, ISCs contributed to their niche after damage and promoted its function in support of crypt regeneration. Rather than simply acting as an alarmin signal for the immune system, IL-33 facilitated dynamic interactions between ISCs and their epithelial niche and orchestrated a controlled

regenerative process following radiation injury. Disturbances in this tightly regulated epithelial crosstalk may contribute to clinical intestinal pathology and disease settings of altered growth, and identification of the IL-33/EGF regenerative axis provides potential targets for therapeutic intervention to inhibit dysregulated proliferation or to promote intestinal recovery after damage.

## Methods

### Mice

B6N.129-Rpl22 (RiboTag) and IL-33-GFP C57BL/6 mice were obtained from Jackson Laboratory. *Olfm4-IRES-eGFPCreERT2*[18], Lyz-DsRed[32], Lyz-Cre[33], Lgr5-LacZ[34], and IL-33 KO[35] C57BL/6 mice were generated as previously described. Mouse maintenance and experimental procedures were performed according to the institutional protocol guidelines and ethical regulations of the Memorial Sloan Kettering Cancer Center (MSKCC) Institutional Animal Care and Use Committee. Mice were housed in micro-isolator cages, up to five per cage, in MSKCC specific-pathogen-free facilities maintained at a room temperature of 72° (+/− 1°) Fahrenheit (~22° Celsius) and a light/dark cycle of 12 h each. Mice received standard chow and autoclaved sterile drinking water. Mice used were male and female with median ages of 2–3 months.

### Crypt isolation and organoid culture

Crypt isolation and organoid culture from WT and IL-33 KO mice were largely performed as described previously[22,36], with some modifications. In brief, after euthanizing the mice with $CO_2$, small intestines were collected, opened longitudinally, and washed with PBS. In most experiments, the tissue collection focused on the distal two-thirds of the small intestine, enriching for ileum, which was the region of small intestine evaluated histologically. Experiments shown in Supplementary Fig. 5 utilized the entire length of the small intestine. To isolate the crypts, the tissue was incubated at 4 °C in EDTA (5 mM) for 30 min. Crypts were then plated in growth-factor-reduced Matrigel (Corning) diluted 50% with DMEM/F12 medium. ENR culture media was prepared as follows: DMEM/F12 (Sigma), 2 mM Glutamax (Invitrogen), 10 mM HEPES (Sigma), 55 μM 2-Mercaptoethanol(Gibco), antibiotic-antimycotic (Gibco), 1 mM *N*-acetyl cysteine (Sigma), N2 supplement (Invitrogen), 50 ng/ml mouse EGF (Peprotech), 50 ng/ml mouse Noggin (Peprotech), and 10% human R-spondin-1 conditioned medium (CM) from R-spondin-1-transfected HEK 293 T cells[37]. In experiments where EGF was omitted from the culture media, this was referred to as NR media. The media was replaced every 2–3 days, and in experiments treating organoids with different concentrations of recombinant mouse IL-33 (R&D Systems), the IL-33 was replaced with the media changes. In experiments treating organoids with mouse ST2-Fc (2 μg/ml), an IL-33 inhibitor, these cultures were compared to treatment with control Fc (2 μg/ml), both from R&D Systems.

For experiments comparing growth properties of WT and IL-33 KO organoids, as shown in Fig. 3c–e, conditions were optimized for organoid culture from epithelial single cells. Fresh crypts were cultured in ENR for 48–72 h in order to generate organoids, which were used for dissociation into single cells by incubating with TrypLE Express (Gibco) for 10 min at 37 °C and then mechanically disrupting with glass Pasteur Pipets. This was followed by several filtration steps with 35 μm strainers, and the resulting single cells were then resuspended in liquid Matrigel and used as the starting material for experiments comparing organoid growth. For experiments testing organoid irradiation, irradiation was performed 24 h after generation of single cells, and cultures were monitored for the subsequent 48 h. For experiments testing organoid passaging, as shown in Fig. 3f, fresh crypts were cultured in NR for five days prior to passaging, then dissociated into single cells as above, resuspended in liquid Matrigel, cultured again in NR, and evaluated five days later. Images of mouse organoids were acquired using a Zeiss Axio Observer.Z1 inverted microscope or a Biotek Cytation 7 imager.

Healthy human ileum organoids were cultured from banked frozen organoids ( > passage 7) that had previously been generated from biopsies obtained during endoscopy of healthy human controls. All healthy controls had been investigated for celiac disease, but ultimately were found to be free of pathologic findings. They provided written informed consent to donate samples for research according to a protocol reviewed and approved by the review board of the UMC Utrecht (METC 10-402/K; TCBio 19-489). Organoids were passaged via single cell dissociation using 1x TrypLE Express (Gibco) and resuspended in medium without growth factors (GF-), comprised of Advanced DMEM/F12 (Gibco), 100 U/ml penicillin-streptomycin (Gibco), 10 mM HEPES (Gibco) and Glutamax (Gibco), and 50–66% Matrigel (Corning). After plating and Matrigel polymerization, human SI organoid expansion medium (hSI EM) was added, consisting of GF-, Wnt CM (50% final concentration), R-spondin CM (20% final concentration), Noggin CM (10% final concentration), 50 ng/ml murine EGF (Peprotech), 10 mM nicotinamide (Sigma), 1.25 mM N-acetylcysteine (Sigma), 2% B27 (Gibco), 500 nM A83-01 (Tocris), and 10 uM SB202190 (Sigma). Medium was refreshed every 2–3 days. For promotion of differentiation into enterocytes, the organoids were cultured in hSI EM for 7 days, and then cultured for 4–5 days in differentiation medium (DM) consisting of hSI EM without Wnt CM, nicotinamide, or SB202190. Images of human organoids were acquired using an EVOS FL Cell Imaging System (Thermo Fisher Scientific).

### IL-33 measurement

IL-33 production was measured using a Quantikine ELISA kit (R&D Systems). The distal two-thirds of the small intestines were collected, enriching for ileum, which was the region of the small intestine evaluated histologically. The tissues were opened longitudinally, and after five rounds of washing in PBS, they were then cut into small segments of 2 mm. Segments were resuspended in 1 ml PBS and homogenized using a Mini-Beadbeater-96 (Biospec). Samples were then spun down at 12,000 g for 5 min, and 50 μl of supernatant was used for the quantitative determination of IL-33. Normalization of IL-33 protein expression was performed using the total protein content determined using the BCA assay. The optical density in each well was measured with Tecan Infinite M1000 pro.

### Flow cytometry

Paneth cells were sorted from crypts isolated from Lyz-DsRed mice. Crypts were isolated using the same protocol described above and then transferred to a gentleMACS C Tube containing 3 ml of digestion buffer [TrypLE, Collagenase D (2 mg/ml) and DNase1 (100 U/ml)] and incubated at 37 °C for 15 min. This was followed by several passages in Corning test tubes with cell strainer snap caps. Cells were isolated with a BD FACSAria III sorter and FACS-sorted directly into ENR media containing Rho-kinase/ROCK inhibitor Y-27632 (20 μM, Tocris Bioscience). Sorted Paneth cells were resuspended in Matrigel (10,000 cells per drop) and challenged with IL-33 (50 ng/ml) or PBS for 24 h in ENR media with Y-27632. Antibodies used for flow cytometry are listed in Supplementary Table 1.

### Organoid measurement

For size evaluation, the surface area of organoid horizontal cross-sections was measured. If all organoids in a well could not be measured, several random non-overlapping pictures were acquired from each well and then analyzed using ImageJ software. Area measurements were automated using the Analyze Particle function of the ImageJ software and validated with the measurement of random fields. For automated size measurements, the threshold for organoid identification was set based on monochrome images. The sizes of the largest and smallest organoids in the reference well were measured manually, and their areas were used as the reference values for setting the minimal and maximal particle sizes. Organoids touching the edge

of the images were excluded from the counting. In order to evaluate growth efficiency, organoid frequencies (numbers per well) were counted manually using brightfield microscopy.

## Histology and immunostaining

For histopathologic analysis of radiation injury, the distal 1–2 inches of the ileum were preserved in 4% paraformaldehyde, paraffin-embedded, and sectioned. Fixed tissue sections were deparaffinized with EZPrep buffer (Ventana Medical Systems) and stained with hematoxylin (Sigma-Aldrich) for 5 min, rinsed in water, dipped quickly in 1% acid ethanol, and washed in water. Slides were then dipped in eosin (Sigma-Aldrich) for 1 min and rinsed in water before dehydration. Coverslips were added with Permount (Fisher Scientific).

For immunostaining, antigen retrieval was performed with CC1 buffer (Ventana Medical Systems). An anti-HA Alexa-488-conjugated antibody was used for HA staining (Biolegend cat. #901509, 1:1000 dilution). Slides were treated for 30 min with 0.2% Triton X-100 and blocked for one hour with 3% BSA prior to staining. Nuclei were counterstained using bisBenzimide H 33342 trihydrochloride (Sigma-Aldrich). Coverslips were added with Vectashield (Vector Laboratories). Lysozyme staining was performed at the Molecular Cytology Core Facility of MSKCC using a Discovery XT processor (Ventana Medical Systems). Sections were blocked for 30 min with Background Buster solution (Innovex) before staining. Anti-lysozyme antibodies (DAKO, 2 μg/ml) were applied, and sections were incubated overnight followed by a 60 min incubation with goat anti-rabbit IgG Alexa 488 (Invitrogen cat. #A11008, 1:800 dilution). For the evaluation of stem cell numbers, small intestines from both WT Lgr5-LacZ and IL-33 KO Lgr5-LacZ mice were collected for histologic analysis and stained with a specific antibody against β-Galactosidase (Invitrogen cat. #11132, 1:1000 dilution). To quantify the number of proliferative cells, crypts were stained with rabbit anti-Ki67 (Cell Signaling Technology cat. #9129, 1:200 dilution).

Slide images were either acquired using an Axio Scan.Z1 scanner (Zeiss) and evaluated using Panoramic Viewer (3DHISTECH) or were acquired using an Axio Observer.Z1 inverted microscope (Zeiss) and then evaluated using ImageJ software.

## Three-dimensional imaging

Mouse small intestines were fixed by paraformaldehyde (4%) perfusion. The fixed tissues were immersed in 2% Triton-X 100 solution for permeabilization. Before the staining steps, tissues were blocked with the blocking solution. Small intestines were then incubated with primary antibody at 1:100 dilution. The primary antibodies used were anti-GFP (Abcam cat. #ab13970, 1:100 dilution), anti-CD45 (eBioscience cat. #14-0451-82, 1:100 dilution), anti-lysozyme (Dako cat. #A009902-2, 1:100 dilution), anti-Olfm4 (Cell Signaling Technology cat. #39141, 1:100 dilution), anti-CD31 (BD Biosciences cat. #553370, 1:100 dilution), and anti-Vimentin (Abcam cat. #ab92547, 1:100 dilution). An Alexa Fluor 647-conjugated goat-anti-rabbit, Alexa Fluor 647-conjugated goat-anti-rat, or Alexa Fluor 488-conjugated goat-anti-chicken secondary antibodies (Invitrogen, 1:500 dilution) was then used to reveal the immunopositive staining. Afterward, tissues were incubated with DiD (4-chlorobenzene sulfonate salt; 2 μg/ml; Invitrogen, D307) to label cellular membranes and DAPI (20 μg/ml, Invitrogen) to label the nuclei. Finally, the labeled specimens were immersed in FocusClear solution (CelExplorer, Hsinchu, Taiwan) for optical clearing before being imaged via confocal microscopy (Zeiss LSM 880). Cellular membrane staining labels all the cells in the tissue, but epithelial cells tend to have stronger membrane labeling than the cells in the lamina propria. To present clear tissue architecture in Fig. 2a, this intrinsic contrast in DiD lipophilic dye staining intensity was further enhanced, providing a better representation of crypt structures within the surrounding mucosa. Amira 6.0.1 image reconstruction software (FEI) was used for processing and projection of the confocal images.

## Immunoblotting analysis

Western blot analysis was carried out on total protein extracts from organoids. Organoids were treated with SB202190 (10 μM, Toscris) for 24 h. Crypts were then transferred to RIPA buffer containing a cocktail of protease and phosphatase inhibitors (Sigma). To homogenize the tissue, samples were sonicated on ice for 20 s (Fisherbrand Model 120 Sonic Dismembrator, Amp 35%). Total protein was determined using the Bicinchoninic Acid Assay Kit (Pierce). Loading 30 μg per lane of lysate, proteins were separated by electrophoresis using a 10% polyacrylamide gel and transferred to nitrocellulose. Membranes were blocked for one hour at room temperature with 1% Blot-Qualified BSA (Promega, cat. #W384A) and 1% non-fat milk (LabScientific, cat. #M0841). They were then incubated overnight at 4 °C with the following primary antibodies: rabbit anti-phospho-p38 (Cell Signaling Technology cat. #4511, 1:1000 dilution) and rabbit anti-p38 (Cell Signaling Technology cat. #8690, 1:1000 dilution), from Cell Signaling Technology. This was followed by incubation with the secondary antibody anti-rabbit HRP (Cell Signaling Technology cat. #7074, 1:1000 dilution) and visualization with the Pierce ECL Western Blotting Substrate (Thermo Scientific, cat. #32106).

## Quantitative PCR analysis

For quantitative (q)PCR, segments of the small intestine or isolated crypts were collected from euthanized mice. Alternatively, RNA was isolated from organoids after in vitro culture. RNA was isolated using the E.Z.N.A. Total RNA Kit from Omega Bio-Tek. Reverse transcription PCR was performed with the High-Capacity RNA-to-cDNA Kit from Applied Biosystems. qPCR was performed using the QuantStudio 7 Flex System (Applied Biosystems) using TaqMan Universal PCR Master Mix (Applied Biosystems) or Power SYBR Green PCR Master Mix. Specific TaqMan primers were obtained from ThermoFisher Scientific: *Lyz1/Lyz2* (Mm00727183_s1); *Gapdh* (Mm99999915_g1). ST2 was measured using isoform-specific primers; mST2: forward 5′-AAGGCACACCATAAGGCTGA-3′ and reverse 5′-TCGTAGAGCTTGCCATCGTT-3′; sST2: forward 5′-TCGAAATGAAAGTTCCAGCA-3′ and reverse 5′-TGTGTGAGGGACACTCCTTAC-3′. The sequences of other primers were obtained from PrimerBank: Gapdh (ID 6679937a1); *Cdk2* (ID 23956072a1); *Lgr5* (ID 6753842a1); *Olfm4* (ID 71892419c1); *Egfr* (ID 10880776a1); *Egf* (ID 6753732a1); Il1rap (ID 6680421a1). Relative amounts of mRNA were calculated by the comparative ΔCt method with Gapdh as a housekeeping gene[38].

For human qPCR, RNA was isolated using TRIzol LS (Thermo Fisher) and stored at −80 °C until further processing. Reverse transcription PCR was performed with the iScript cDNA Synthesis Kit (BioRad). Primers were obtained from Integrated DNA Technologies following primer design with the NCBI Nucleotide and Primer Blast databases; *LGR5*: forward 5′-GAATCCCCTGCCCAGTCTC-3′ and reverse 5′-TCTTAAACGCTTCGGAAGTTA-3′; *HP1BP3*: forward 5′-CCCACGTCCCAAGATGGAT-3′ and reverse 5′-AAGGTCTTCTCACCACGTAGTC-3′; *IL33*: forward 5′-GTGACGGTGTTGATGGTAAGAT-3′ and reverse 5′-AGCTCCACAGAGTGTTCCTTG-3′. cDNA was amplified with iQ SYBR Green Supermix (BioRad) in a CFX96TM Real-Time PCR Detection System (BioRad). Relative amounts of mRNA were calculated by the comparative ΔCt method with HP1BP3 as housekeeping gene.

## Tamoxifen treatment and ribosome immunoprecipitation

Stem-cell-specific (Olfm4-Ribo) and Paneth-cell-specific (Lyz-Ribo) mice were obtained by crossing B6N.129-*Rpl22* RiboTag mice (Jackson Laboratories) with *Olfm4-IRES-eGFPCreERT2* knockin mice or Lyz-Cre mice generated by H. Clevers. Prior to administration for induction of gene recombination, tamoxifen (20 mg/ml) was dissolved in warm corn oil (Sigma). Corn oil vehicle alone was administered to control

groups. Olfm4-Ribo mice were treated with a single intraperitoneal injection of tamoxifen (80 mg/kg). Mice were examined 20–24 h following treatment. Lyz-Ribo mice received a single injection of tamoxifen (80 mg/kg) for two consecutive days before animals were examined. SI was extracted from mice, and SI homogenates were obtained as follows: 2 mg of distal SI from each animal was homogenized in cold immunoprecipitation (IP) lysis Buffer (Thermo Fisher) supplemented with 200 U/ml RNAsin (Promega), 1x protease inhibitor cocktail (Sigma-Aldrich), and 500 μg/mL cycloheximide. Stainless steel beads measuring 2.3 mm in diameter (BioSpec) were added to the solution, and the tissue was homogenized for one minute using Mini-Beadbeater-96 (Biospec). In order to remove cell debris, samples were spun down at 10,000 x g for 5 min, and the supernatants were collected.

For IP, 50 μl of Dynabeads Protein G (Thermo Fisher) were coupled with 10 μg of anti-HA antibody (Pierce). After 30 min of incubation at room temperature, beads coupled with anti-HA antibodies were washed in PBS with 0.05% Tween 20 by gently pipetting and then added to each sample and incubated with rotation overnight at 4 °C. The following day, tubes were placed in a magnet and washed twice with IP buffer supplemented with 500 μg/mL cycloheximide. To synthesize cDNA from the total RNA recovered after IP, 1:2 volumes of RNA-Solv (Omega Bio-Tek) were added to the pellets consisting of bead-Ab-Ag complexes. Total RNA was isolated using the E.Z.N.A Total RNA kit from Omega Bio-Tek according to the manufacturer's protocol. Total RNA was measured using a nanodrop 1000 (Thermo Fisher) and converted to cDNA using the High Capacity RNA-to-cDNA Kit from Applied Biosystems. No detectable RNAs were pulled down from the control group, suggesting there was little to no non-specific binding from anti-HA or protein G magnetic beads during the IP.

### In vivo EGF and gefitinib administration

For EGF administration, recombinant mouse EGF was purchased from Peprotech and reconstituted as described by the manufacturer to a concentration of 1 mg/mL in PBS. Mice at eight weeks of age were treated daily by intraperitoneal injection with either PBS alone or PBS containing 10 μg EGF. EGF administration was started on the same day as irradiation (10 Gy TBI) and continued for an additional two days. For EGFR inhibition, gefitinib was freshly prepared in DMSO (40 mg/mL) prior to administration by oral gavage. Control groups were treated with vehicle (PBS, 0.5% methylcellulose, and 0.5%, polysorbate 80). The daily dose of gefitinib was 1 mg/mouse ( ~ 50 mg/kg body weight), administered once daily starting from the day of TBI (10 Gy) until two days after.

### Statistics and software

All experiments were performed at least twice, unless otherwise specified, and no mice were excluded from experimental analyses. A priori power analysis was not performed. All tests performed were two-sided. For the comparisons of two groups, a *t*-test or non-parametric test was performed. ANOVA was utilized for analysis of experiments with multiple comparisons. In most cases, non-parametric testing was performed if normal distribution could not be assumed. qPCR reactions and ordinal outcome variables were tested non-parametrically. Statistical analyses of organoid sizes were based on all evaluable organoids in the visual field, typically more than 25 per group. Statistical analyses of organoid numbers and efficiency were based on individual wells. All in vitro experiments included material coming from at least two different mice. Statistical analyses of in vivo stem cell numbers in Lgr5-LacZ mice were performed on several independent sections from multiple mice. Statistics were calculated and graphs were generated using Graphpad Prism version 8 (GraphPad Software, San Diego, CA).

### Reporting summary

Further information on research design is available in the Nature Portfolio Reporting Summary linked to this article.

## Data availability

Graphical data generated in this study, including data presented as bar graphs, line graphs, or overlapping data points, have been included as raw data in the Source Data file. The uncropped western blot shown in Fig. 3h has been include in the Source Data file as well. Source data are provided with this paper.

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

## Acknowledgements

We thank Jarrod A. Dudakov, Robert R. Jenq, and Karuna Ganesh for their valuable advice, and we gratefully acknowledge the technical assistance of the MSKCC Research Animal Resource Center and Molecular Cytology Core Facility. We also thank the Integrated Genomics Operation Core, funded by MSKCC's NCI Cancer Center Support Grant (CCSG, P30-CA08748), Cycle for Survival and the Marie-Josée and Henry R. Kravis Center for Molecular Oncology. This research was supported by National Institutes of Health awards R01-HL125571 (A.M.H), R01-HL146338 (A.M.H), R01-HL145631 (A.M.H), R01-HL56067 (B.R.B), R37-AI34495 (B.R.B), R01-HL11879 (B.R.B), P01-CA65493 (B.R.B), R21-AI121981 (H.R.T), R56-AI139327 (H.R.T), R01-HL122489 (H.R.T), and P30-CA008748 (MSKCC Core Grant). Support was also received from the Susan and Peter Solomon Divisional Genomics Program, the Ludwig Center for Cancer Immunotherapy, the Parker Institute for Cancer Immunotherapy, and the Anna Fuller Fund (A.M.H.). S.T. was supported by a scholarship from the Mochida Memorial Foundation for Medical and Pharmaceutical Research, an American Society for Blood and Marrow Transplantation (ASBMT, now ASTCT) New Investigator Award from Millenium, the Takeda Oncology Company, and a John Hansen Research Grant from DKMS. Y.F. was also supported by an ASBMT (now ASTCT) New Investigator award and by an Amy Strelzer Manasevit Resreach program Fellowship Award. P.V. was supported by the American Italian Cancer Foundation and an ASBMT (now ASTCT) New Investigator Award. V.A. was supported by the German Research Foundation (DFG).

## Author contributions

M.C. designed and performed most experiments, Y.F. performed three-dimensional imaging analyses, and P.V. performed in vivo gefitinib treatment and Thy 1.2 depletion experiments. A.E. and J.K. maintained the mouse colonies, S.J. and C.L. conducted human organoid culture experiments, and S.N., J.v.E., and H.C. generated transgenic mice. V.A., W.C., S.T., T.I., J.S., and H.T. contributed to study design and interpretation, and B.R.B. and A.M.H. supervised the research. All authors were included in manuscript editing.

## Competing interests

B.R.B. receives remuneration as an advisor to Kamon Pharmaceuticals, Inc, Five Prime Therapeutics Inc, Regeneron Pharmaceuticals, Magenta Therapeutics, and BlueRock Therapeuetics, research support from Fate Therapeutics, RXi Pharmaceuticals, Alpine Immune Sciences, Inc, Abbvie Inc., Leukemia and Lymphoma Society, Childrens' Cancer Research Fund, and KidsFirst Fund, and is a co-founder of Tmunity. A.M.H. and C.A.L. hold intellectual property related to Interleukin-22, and A.M.H. has a collaboration with Evive Biotechnology (Shanghai) Ltd, which supported a multicenter clinical trial studying use of Interleukin-22 in patients with graft vs. host disease. A.M.H. also serves in a volunteer capacity as a member of the Board of Directors of the American Society for Transplantation and Cellular Therapy (ASTCT). The remaining authors declare no competing interests.
