## [Peer Review File · Nature Communications]

REVIEWERS' COMMENTS

Reviewer #1 (Remarks to the Author):

The authors have addressed my points raised in the previous review.

[Redacted]

Reviewer #2 (Remarks to the Author):

I am supportive of publication of this study. The key point was the discrepancy between the in vitro and in vivo work and the epithelial centric focus. Although because of these concerns, in my opinion, it falls short for consideration by Nature I think it is a valuable addition to the field and Nature communications a good outlet. The data that have been added, and more importantly the restructuring of the manuscript, have further contributed to the relevance of this study. The key issues by reviewer #2 and #3 have been addressed.

Reviewer #3 (Remarks to the Author):

Califiore et al Manuscript NCOMMS-23-11077-T

Reviewer comments:

Since this reviewers last review of this manuscript, the authors have done a truly admirable and enormous amount of additional experiments and work reflected in multiple new figures and extended figures to address the issues brought up in the previous review 3 years ago by multiple reviewers. The authors have also impressively included costly and difficult experiments [Redacted]. They have cleaned up the Materials and Methods Section that now reads very well. The authors have addressed this

reviewers concerns about the sources and location of IL-33 expression and have done a huge amount of work to address the issue of a potentially homeostatic role for IL-33 signaling in maintenance of the ileal intestinal crypt. The new 2D and 3D confocal microscopy data presented in Figure 2c-f are extremely enlightening with respect to cell-type-specific expression of IL-33 within the crypt in this TBI model. They have also greatly improved and developed the ex vivo small intestine organoid model and made this model more relevant to the in vivo TBI model.

This reviewer also appreciates that the figures and figure legends, as in the previous submission in 2019, allow the reader to fully evaluate and interpret the data without having to refer to the text constantly to understand the point of the figure, and the figures and figure legends are superbly developed.

Specific Comments:

In future studies, it would be of great interest to know if there are significant differences in the radiation injury IL-33/sST2/EGF/EGFR mouse model if you move the mouse model from the C57BL/6 onto different genetic backgrounds (129sv; CD-1; CF-1); as has been known now for a few decades, and exemplified by the original EGFR KO mouse model, that mouse inbred backgrounds can demonstrate a significant effect on the biology all the way from embryonic lethal phenotypes to live pups with specific deficits in various tissues where EGFR is expressed (Threadgill DW, Dlugosz AA, Hansen LA, Tennenbaum T, Lichti U, Yee D, LaMantia C, Mourton T, Herrup K, Harris RC, et al. Targeted disruption of mouse EGF receptor: effect of genetic background on mutant phenotype. *Science*. 1995 Jul 14;269(5221):230-4. doi: 10.1126/science.7618084. PMID: 7618084; Sibilgia M, Wagner EF. Strain-dependent epithelial defects in mice lacking the EGF receptor. *Science*. 1995 Jul 14;269(5221):234-8. doi: 10.1126/science.7618085. Erratum in: *Science* 1995 Aug 18;269(5226):909. PMID: 7618085; Miettinen PJ, Berger JE, Meneses J, Phung Y, Pedersen RA, Werb Z, Derynck R. Epithelial immaturity and multiorgan failure in mice lacking epidermal growth factor receptor. *Nature*. 1995 Jul 27;376(6538):337-41. doi: 10.1038/376337a0. PMID: 7630400). As stated by Sibilgia et al 1995, "These results indicate that the EGFR regulates epithelial proliferation and differentiation and that the genetic background influences the resulting phenotype."

Furthermore, in future studies it would be significant to explore the organoid culture experiments presented in Figure 3 and Extended Figure 4 and 5 on different mouse genetic backgrounds, given what has been observed in EGFR KO mice on different backgrounds, in order to see what type of variations exist in IL-33 production, EGF and EGFR responsiveness that may be mediated by genetic modifiers provided by these different genetic backgrounds.

It is remarkable that such a small increase in IL-33 protein as indicated in Extended Figure 1b from 1.0 ng/mg to 1.5 ng/mg (so a 50% increase) results from the large increase in mRNA for IL-33 (relative 1.0 to 3.0, so a 3-fold change) can result in such a large change in ISC and Paneth cell expansion. Furthermore, the question of a homeostatic role for IL-33 in the intestinal crypt environment is still a relevant

question as there does appear to be a trend of decreased Paneth cell numbers in the IL-33 KO Extended Figure 1e. In Figure 1j and 1k there also appears to be a similar trend of fewer ISCs (albeit not statistically significant). So there may indeed be some reduction in ISCs and Paneth cells in the IL-33 in the unirradiated state between WT and IL-33 KO.

The demonstrated increased dependence of IL-33 KO organoid cultures on EGF in the absence of IL-33 (IL-33 KO) as demonstrated in Figure 3 also appears to support a homeostatic role for IL-33 production in the intestinal crypt (at least in this SI organoid model), although Figure 3c and 3d appear to suggest that WT and IL-33 unirradiated organoids have the same area and organoids per well, but it is assumed that this is in ENR media containing exogenous EGF, as opposed to Extended Figure 3e-j where NR media is lacking in exogenous EGF, and an increased dependence on EGF is revealed in both the unirradiated and irradiated state. This increased dependence on EGF in the organoid model is not found in the in vivo model, as in the unirradiated state WT and IL-33 KO crypt circumference, crypt depth, Ki67+ cells per crypt and Lgr5+ ISC cells are equivalent as demonstrated in Figure 1c-k. Perhaps this reviewer is not interpreting Figure 3 correctly in terms of perceiving the increased dependence of IL-33 KO organoids on EGF as existing also in the unirradiated or homeostatic state?

In Figure 1c is there an artifact of a thinner muscularis propria in the irradiated IL-33 KO animal? Or is this an additional phenotype seen in the irradiated IL-33 KO mouse? It appears from the section presented quite dramatic.

In Extended Figure 1e it would be best to add a bar similar to Extended Figure 1d that indicates Day 3 as this reviewer was confused when first inspecting the Figure before reading the text.

The additional and extensive 2D and 3D confocal microscopy work done since 2019 in Figure 2 and Extended Figure 2 has helped to clarify the sources of IL-33 production after irradiation, and clearly demonstrate that within the intestinal crypt there is a unique source of IL-33 production locally that coincides with the Olfm4+ ISC cell type and not CD31+ hematopoietic, stromal vimentin+ fibroblast, CD45+ lymphocyte and other cell types in this location. And this is further supported by Extended Figure 3 that examines the contribution of lymphocytes such as CD45+ CD4+ T cells and Group 2 and 3 ILCs. However, in Figure 2d and 2e there does appear to be significant extracrypt IL-33 GFP staining both before and 48 hours post TBI—This is assumed to be vimentin+ intercrypt stromal fibroblasts producing IL-33 not located in the crypts as outlined in Extended Figure 2e. But could this also be lymphocytes such as CD45+ CD4+ T cells, ISC2 or ISC3 lymphocytes in addition, although that is not apparently supported by Figure 2c? It is potentially surprising that these immune cells do not produce IL-33 in the extracrypt region in response to radiation injury, even though they do not appear to be localized to the intestinal crypts and do not express IL-33 within the crypts as per Figure 2c. Given all the careful FACS sorting on CD45+ lymphocytes from the mouse small intestine in Extended Figure 3, it might have been worthwhile to demonstrate that this cell type that is located near the intestinal crypt can indeed produce IL-33 although perhaps not from TBI. The overall question is since there is extracrypt expression of IL-33 in

vimentin+ stromal fibroblast and perhaps also extracrypt CD45+ lymphocytes do they contribute in any meaningful way to IL-33 mediated intestinal crypt regeneration, and to what extent do they contribute to the total amount of IL-33 mRNA and protein measured in Figure 1a and Extended Figure 1a and 1b. Furthermore, is this reviewer wrong (perhaps) in observing in Figure 2c that IL-33 expression in the intercrypt region is greater 48 hours post-TBI?

Extended Figure 3g: This reviewer believes the Y axis should be in micrometers (μM), not millimeters (mM), as it is impossible to imagine average crypt sizes of 140mM (14 centimeters) and assume this is a simple mistake.

In Materials and Methods under Quantitative PCR Analysis it would be best to add 5'- and 3'- to each primer set listed as nucleotides just to avoid any potential mistakes for individuals wishing to replicate or use these primers from other laboratories. Also, as Dll1 and Dll4 do not appear in any of the data presented including the Extended Figures, these should be removed from the Materials and Methods under Quantitative PCR Analysis.

Also under Quantitative PCR Analysis "Relative amounts of mRNA were calculated by the comparative ΔCt method with Gapdh as a housekeeping gene." This should have a reference.

Under Immunoblotting Analysis "Fisherbrand Model 120 sonic dismembrator," "Sonic Dismembrator" should be capitalized.

Response for manuscript NCOMMS-23-11077-T

REVIEWERS' COMMENTS

Reviewer #1 (Remarks to the Author):

The authors have addressed my points raised in the previous review.

[Redacted]

We thank the reviewer for their positive assessment.

Reviewer #2 (Remarks to the Author):

I am supportive of publication of this study. The key point was the discrepancy between the in vitro and in vivo work and the epithelial centric focus. Although because of these concerns, in my opinion, it falls short for consideration by Nature I think it is a valuable addition to the field and Nature communications a good outlet. The data that have been added, and more importantly the restructuring of the manuscript, have further contributed to the relevance of this study. The key issues by reviewer #2 and #3 have been addressed.

We thank the reviewer for their support of the revised manuscript, which we agree has been greatly improved through this review process.

Reviewer #3 (Remarks to the Author):

Califiore et al Manuscript NCOMMS-23-11077-T

Reviewer comments:

Since this reviewers last review of this manuscript, the authors have done a truly admirable and enormous amount of additional experiments and work reflected in multiple new figures and extended figures to address the issues brought up in the previous review 3 years ago by multiple reviewers. The authors have also impressively included costly and difficult experiments [Redacted]. They have cleaned up the Materials and Methods Section that now reads very well. The authors have addressed this reviewers concerns about the sources and location of IL-33 expression and have done a huge amount of work to address the issue of a potentially homeostatic role for IL-33 signaling in maintenance of the ileal intestinal crypt. The new 2D and 3D confocal microscopy data presented in Figure 2c-f are extremely enlightening with respect to cell-type-specific expression of IL-33 within the crypt in this TBI model. They have also greatly improved and developed the ex vivo small intestine organoid model and made this model more relevant to the in vivo TBI model.

This reviewer also appreciates that the figures and figure legends, as in the previous submission in 2019, allow the reader to fully evaluate and interpret the data without having to refer to the text constantly to understand the point of the figure, and the figures and figure legends are superbly developed.

We thank the reviewer for the complimentary assessment of the revised manuscript and

the substantial amount of effort that went into it.

Specific Comments:

In future studies, it would be of great interest to know if there are significant differences in the radiation injury IL-33/sST2/EGF/EGFR mouse model if you move the mouse model from the C57BL/6 onto different genetic backgrounds (129sv; CD-1; CF-1); as has been known now for a few decades, and exemplified by the original EGFR KO mouse model, that mouse inbred backgrounds can demonstrate a significant effect on the biology all the way from embryonic lethal phenotypes to live pups with specific deficits in various tissues where EGFR is expressed (Threadgill DW, Dlugosz AA, Hansen LA, Tennenbaum T, Lichti U, Yee D, LaMantia C, Mourton T, Herrup K, Harris RC, et al. Targeted disruption of mouse EGF receptor: effect of genetic background on mutant phenotype. *Science*. 1995 Jul 14;269(5221):230-4. doi: 10.1126/science.7618084. PMID: 7618084; Sibia M, Wagner EF. Strain-dependent epithelial defects in mice lacking the EGF receptor. *Science*. 1995 Jul 14;269(5221):234-8. doi: 10.1126/science.7618085. Erratum in: *Science* 1995 Aug 18;269(5226):909. PMID: 7618085; Miettinen PJ, Berger JE, Meneses J, Phung Y, Pedersen RA, Werb Z, Derynck R. Epithelial immaturity and multiorgan failure in mice lacking epidermal growth factor receptor. *Nature*. 1995 Jul 27;376(6538):337-41. doi: 10.1038/376337a0. PMID: 7630400). As stated by Sibia et al 1995, "These results indicate that the EGFR regulates epithelial proliferation and differentiation and that the genetic background influences the resulting phenotype."

Furthermore, in future studies it would be significant to explore the organoid culture experiments presented in Figure 3 and Extended Figure 4 and 5 on different mouse genetic backgrounds, given what has been observed in EGFR KO mice on different backgrounds, in order to see what type of variations exist in IL-33 production, EGF and EGFR responsiveness that may be mediated by genetic modifiers provided by these different genetic backgrounds.

We thank the reviewer for this comment. We share the sentiment and agree that examining the pathway in more detail in mice with different genetic backgrounds and further focusing on EGFR will be an interesting and important future direction for us.

It is remarkable that such a small increase in IL-33 protein as indicated in Extended Figure 1b from 1.0 ng/mg to 1.5 ng/mg (so a 50% increase) results from the large increase in mRNA for IL-33 (relative 1.0 to 3.0, so a 3-fold change) can result in such a large change in ISC and Paneth cell expansion. Furthermore, the question of a homeostatic role for IL-33 in the intestinal crypt environment is still a relevant question as there does appear to be a trend of decreased Paneth cell numbers in the IL-33 KO Extended Figure 1e. In Figure 1j and 1k there also appears to be a similar trend of fewer ISCs (albeit not statistically significant). So there may indeed be some reduction in ISCs and Paneth cells in the IL-33 in the unirradiated state between WT and IL-33 KO.

We agree with the reviewer, but it is possible that overall measured cytokine concentrations may differ from concentrations in localized microenvironments, and it is difficult to know how much signal individual cells are exposed to within a tissue. Additionally, we also noticed the possible trend in reduction of Paneth cell frequencies and ISC gene expression in homeostasis in IL-33 KO mice, which could be directly related, however the differences were not statistically significant and were not recapitulated at the cellular level in ISCs in homeostasis (ISC frequency, Fig. 1h-i). In addition, in the absence of damage the overall crypt frequency and morphology did not appear affected in the IL-33 KO mice (Fig. 1d-e). Therefore, the biologic significance of these homeostatic observations is unclear. Perhaps during homeostasis there could be more subtle insults to the tissue that arise and invoke this pathway without rising to the level of overt significance as observed in dedicated injury models such as irradiation. Such subtle insults are of course more

challenging to study, but they could certainly be relevant for providing a role for the IL-33 pathway in homeostasis.

The demonstrated increased dependence of IL-33 KO organoid cultures on EGF in the absence of IL-33 (IL-33 KO) as demonstrated in Figure 3 also appears to support a homeostatic role for IL-33 production in the intestinal crypt (at least in this SI organoid model), although Figure 3c and 3d appear to suggest that WT and IL-33 unirradiated organoids have the same area and organoids per well, but it is assumed that this is in ENR media containing exogenous EGF, as opposed to Extended Figure 3e-j where NR media is lacking in exogenous EGF, and an increased dependence on EGF is revealed in both the unirradiated and irradiated state. This increased dependence on EGF in the organoid model is not found in the *in vivo* model, as in the unirradiated state WT and IL-33 KO crypt circumference, crypt depth, Ki67+ cells per crypt and Lgr5+ ISC cells are equivalent as demonstrated in Figure 1c-k. Perhaps this reviewer is not interpreting Figure 3 correctly in terms of perceiving the increased dependence of IL-33 KO organoids on EGF as existing also in the unirradiated or homeostatic state?

We consider the organoid model, where individual crypts or cells are disrupted and stimulated with potentially high concentrations of growth factors, to represent more of a regeneration model than a model of homeostatic epithelial maintenance. As such, we interpret the findings in Figure 3 to reflect an increased dependence of IL-33 KO organoids on exogenous EGF that does exist even in an unirradiated state, but not necessarily in a homeostatic state. While our study provides insight into the upregulation of EGF in the ISC niche after damage, expression of EGF appeared relatively intact *in vivo* in unirradiated IL-33 KO mice (Fig. 4a), and interpretations of basal EGF regulation in homeostasis are likely beyond the scope of this study.

In Figure 1c is there an artifact of a thinner muscularis propria in the irradiated IL-33 KO animal? Or is this an additional phenotype seen in the irradiated IL-33 KO mouse? It appears from the section presented quite dramatic.

In response to this question, we have gone back and examined the muscularis layer in multiple histologic sections from different IL-33 KO mice after radiation injury. They all appeared to have a thinner layer of muscularis propria, so we do not believe that this represents a tissue processing artifact. None of the sections from mice in the other three experimental conditions processed at the same time demonstrated this phenotype. However, we do not have any specific data investigating or supporting a role for IL-33 in directly impacting the muscle layer. Perhaps the reduced thickness could result from changes in the epithelial layer or from some other indirect aspect downstream of IL-33 biology, or perhaps something direct that is independent of the cells and pathways we have studied here. As such, we feel that it is more appropriate to not emphasize this potential observation.

In Extended Figure 1e it would be best to add a bar similar to Extended Figure 1d that indicates Day 3 as this reviewer was confused when first inspecting the Figure before reading the text.

Done.

The additional and extensive 2D and 3D confocal microscopy work done since 2019 in Figure 2 and Extended Figure 2 has helped to clarify the sources of IL-33 production after irradiation, and clearly demonstrate that within the intestinal crypt there is a unique source of IL-33 production locally that

coincides with the Olfm4+ ISC cell type and not CD31+ hematopoietic, stromal vimentin+ fibroblast, CD45+ lymphocyte and other cell types in this location. And this is further supported by Extended Figure 3 that examines the contribution of lymphocytes such as CD45+ CD4+ T cells and Group 2 and 3 ILCs. However, in Figure 2d and 2e there does appear to be significant extracrypt IL-33 GFP staining both before and 48 hours post TBI—This is assumed to be vimentin+ intercrypt stromal fibroblasts producing IL-33 not located in the crypts as outlined in Extended Figure 2e. But could this also be lymphocytes such as CD45+ CD4+ T cells, ISC2 or ISC3 lymphocytes in addition, although that is not apparently supported by Figure 2c? It is potentially surprising that these immune cells do not produce IL-33 in the extracrypt region in response to radiation injury, even though they do not appear to be localized to the intestinal crypts and do not express IL-33 within the crypts as per Figure 2c. Given all the careful FACS sorting on CD45+ lymphocytes from the mouse small intestine in Extended Figure 3, it might have been worthwhile to demonstrate that this cell type that is located near the intestinal crypt can indeed produce IL-33 although perhaps not from TBI. The overall question is since there is extracrypt expression of IL-33 in vimentin+ stromal fibroblast and perhaps also extracrypt CD45+ lymphocytes do they contribute in any meaningful way to IL-33 mediated intestinal crypt regeneration, and to what extent do they contribute to the total amount of IL-33 mRNA and protein measured in Figure 1a and Extended Figure 1a and 1b. Furthermore, is this reviewer wrong (perhaps) in observing in Figure 2c that IL-33 expression in the intercrypt region is greater 48 hours post-TBI?

IL-33 is not typically produced by lymphocytes (Liew, Girard, and Turnquist. Interleukin-33 in health and disease. *Nature Reviews Immunology* 2016), but given the new biology we have identified here, we felt that it was important to evaluate this experimentally in our model. As shown in Fig. 2c we formally tested if intestinal CD45⁺ lymphocytes could be a source of IL-33 at baseline or after irradiation. Using this 3-D imaging approach, we could reliably identify lymphocytes as either intraepithelial or extracryptal lamina propria lymphocytes, and we could further identify if lymphocytes in either compartment were IL-33⁺. However, we could not identify a population of IL-33-expressing lymphocytes in either setting. Given that this was consistent with the known literature, we did not pursue this possibility further. While we do not believe that lymphocytes are an important source of intestinal IL-33, it is still possible that they could contribute to IL-33-dependent crypt regeneration by being targets of IL-33 that promote regeneration. Despite this theoretical possibility, the data presented in Extended Data Figure 3 (now Supplementary Figure 3) and further supported by the epithelial culture model shown in Figure 3 (in the main text) indicate that IL-33 can promote crypt regeneration independent of its immunomodulatory properties and without any contribution from lymphocytes. This direct effect in the epithelium, and in particular the effect on Paneth cells, thus became a major focus of this study. We nonetheless agree with the reviewer, and we are interested in examining the potential roles of stromal-derived IL-33 and of IL-33-modulated lymphocytes in epithelial regeneration in our future directions.

Extended Figure 3g: This reviewer believes the Y axis should be in micrometers (uM), not millimeters (mM), as it is impossible to imagine average crypt sizes of 140mM (14 centimeters) and assume this is a simple mistake.

Thank you for catching this. It has now been fixed.

In Materials and Methods under Quantitative PCR Analysis it would be best to add 5'- and 3'- to each primer set listed as nucleotides just to avoid any potential mistakes for individuals wishing to replicate or use these primers from other laboratories. Also, as Dll1 and Dll4 do not appear in any of the data

presented including the Extended Figures, these should be removed from the Materials and Methods under Quantitative PCR Analysis.

Done.

Also under Quantitative PCR Analysis “Relative amounts of mRNA were calculated by the comparative ΔC_t method with Gapdh as a housekeeping gene.” This should have a reference.

Done.

Under Immunoblotting Analysis “Fisherbrand Model 120 sonic dismembrator,” “Sonic Dismembrator” should be capitalized.

This has been fixed.